# Unravelling the MicroRNA-Mediated Gene Regulation in Developing Pongamia Seeds by High-Throughput Small RNA Profiling

**DOI:** 10.3390/ijms20143509

**Published:** 2019-07-17

**Authors:** Ye Jin, Lin Liu, Xuehong Hao, David E. Harry, Yizhi Zheng, Tengbo Huang, Jianzi Huang

**Affiliations:** 1Guangdong Key Laboratory of Plant Epigenetics, College of Life Sciences and Oceanography, Shenzhen University, Shenzhen 518060, China; 2TerViva, Oakland, CA 94612, USA

**Keywords:** *Millettia pinnata*, woody oilseed plants, seed development, miRNA

## Abstract

Pongamia (*Millettia pinnata* syn. *Pongamia pinnata*) is a multipurpose biofuel tree which can withstand a variety of abiotic stresses. Commercial applications of Pongamia trees may substantially benefit from improvements in their oil-seed productivity, which is governed by complex regulatory mechanisms underlying seed development. MicroRNAs (miRNAs) are important molecular regulators of plant development, while relatively little is known about their roles in seed development, especially for woody plants. In this study, we identified 236 conserved miRNAs within 49 families and 143 novel miRNAs via deep sequencing of Pongamia seeds sampled at three developmental phases. For these miRNAs, 1327 target genes were computationally predicted. Furthermore, 115 differentially expressed miRNAs (DEmiRs) between successive developmental phases were sorted out. The DEmiR-targeted genes were preferentially enriched in the functional categories associated with DNA damage repair and photosynthesis. The combined analyses of expression profiles for DEmiRs and functional annotations for their target genes revealed the involvements of both conserved and novel miRNA-target modules in Pongamia seed development. Quantitative Real-Time PCR validated the expression changes of 15 DEmiRs as well as the opposite expression changes of six targets. These results provide valuable miRNA candidates for further functional characterization and breeding practice in Pongamia and other oilseed plants.

## 1. Introduction

Woody oilseed plants have attracted increasing attention as alternative feedstocks for biodiesel production in recent years [1]. While conventional feedstocks such as soybean (*Glycine max*), rapeseed (*Brassica napus*), and maize (*Zea mays*) are mainly herbaceous crops that need to be grown on arable lands, the woody oilseed plants are usually adapted to more diverse environments such as mountain areas or intertidal zones [2]. Pongamia is one such tree species with high yield of non-edible seed oils that are highly suitable for biodiesel preparation [3,4]. Pongamia trees can not only withstand a variety of adverse conditions like drought and high salinity [5,6], but also undergo symbiotic nitrogen fixation by their root nodules [7,8]. Hence, they can be planted on wastelands or marginal lands with limited impact on food production and reduced consumption of nitrogen fertilizers. In addition, the extracts and derivatives from various parts of the plant have shown certain pharmacological activities such as antioxidant, antimicrobial, antidiabetic, and antihyperammonemic [9,10,11], which may potentially contribute to increasing the added values of the biodiesel products from this species.

To establish large-scale plantations of Pongamia trees for commercial biodiesel production, one of the major requirements is the improvement in oil-seed productivity, which requires a thorough understanding of the regulatory mechanisms underlying seed development in this species. Tissue- and stage-specific gene expression patterns related to seed development have already been revealed in major oilseed crops through global transcriptome profiling [12,13,14]. In Pongamia, several efforts have developed a bulk of transcriptomic data as a preliminary attempt on characterizing functional genes and profiling their expression at transcriptional level [6,15,16,17]. These transcriptomic data, along with the reference gene sequences from soybean, have facilitated the isolation and characterization of four circadian clock genes (*ELF4*, *LCL1*, *PRR7*, and *TOC1*) and two fatty acid desaturase genes (*SAD* and *FAD2*), which have shown distinct patterns of transcriptional regulation in relation to flowering time and seed development in Pongamia [18,19,20].

On the other hand, microRNAs (miRNAs) are a class of non-coding small RNAs that mainly regulate gene expression at post-transcriptional level [21,22]. As the upstream regulators of protein-coding genes, miRNAs participate in a wide range of biological processes essential for plant growth and stress responses [23,24]. With regard to seed development, miRNA-mediated regulations have already been uncovered at early or late developmental stages in Arabidopsis. For instance, miR156 can target *SQUAMOSA promoter-binding protein-like* (*SPL*) *10* and *SPL11* during early embryogenesis to repress precocious accumulation of certain transcripts normally expressed in maturation phase [25], while miR160-directed repression of *Auxin Response Factor* (*ARF*) *10* plays important roles in seed germination and post-embryonic developmental programs [26]. In addition, with the rapid development of high-throughput sequencing technology, numerous conserved and novel miRNAs have also been identified in developing seeds of a number of oilseed crops, such as rapeseed [27,28], soybean [29,30], and peanut (*Arachis hypogaea*) [31,32]. These studies have implicated the regulatory roles of miRNAs in multiple steps of seed development with special interests in seed oil accumulation. Nevertheless, to the best of our knowledge, large-scale identifications of miRNAs have only been reported in a few oilseed tree species [33,34].

In our previous studies, we have characterized three major developmental phases (i.e., embryogenesis phase, seed-filling phase, and desiccation phase) of Pongamia seeds based on morphological changes and physiological events [35]. In this work, we presented the first comprehensive investigation of miRNAs in Pongamia seeds through Illumina sequencing of nine small RNA libraries from these three phases. Millions of small RNA reads were generated, leading to the discovery of hundreds of conserved and novel miRNAs, among which a subset of miRNAs were identified with differential expressions between developmental phases. Furthermore, putative target genes for these miRNAs were obtained by computational prediction. Meanwhile, expression patterns of several selected miRNAs and their predicted targets during seed development were also examined by quantitative Real-Time PCR (qRT-PCR). Lastly, the potential roles of miRNAs and their target genes in Pongamia seed development were discussed.

## 2. Results

### 2.1. Deep Sequencing of Small RNA Libraries in Developing Pongamia Seeds

To characterize miRNAs and their expression profiles in developing Pongamia seeds, we constructed nine small RNA libraries based on the seed samples collected at the three developmental phases, each with three biological replicates. A total of 283,250,572 raw reads were yielded from the nine libraries by Illumina platform (Table 1). After filtering out low-quality sequences, adapter contaminants, polyA-containing sequences, and reads smaller than 18 nt, approximately 18 to 21 million clean reads were obtained for each library. These clean reads accounted for 62.22% to 67.49% of the raw reads and represented 2.44 to 3.43 million unique sequences in each library. The length distribution of these small RNA reads showed that 24-nt RNAs were the largest population in all nine libraries from the three developmental phases, while the second most abundant group was 21-nt RNAs in the libraries from the embryogenesis phase, but was 22-nt RNAs in the libraries from both the seed-filling phase and the desiccation phase (Appendix A). Subsequently, the clean reads were searched against the Rfam and the NCBI GenBank databases to identify rRNA, tRNA, snRNA, and snoRNA sequences. According to the combined results from both databases, the percentage of these four types of non-coding small RNAs was substantially higher in libraries from the embryogenesis phase (8.31 ± 0.77% of the clean reads) than in those from the seed-filling phase (3.97 ± 0.49%) and the desiccation phase (5.85 ± 0.68%) (Appendix A). For further analysis, the clean reads matching the above four types of small RNAs were excluded. In general, both the length distribution and the relative abundance of various types of small RNAs exhibited a developmental phase-specific pattern in Pongamia seeds.

### 2.2. Identification of Conserved and Novel miRNAs in Developing Pongamia Seeds

To identify miRNA homologs in Pongamia seeds, we used the remaining clean reads in each library to match against the Viridiplantae mature miRNAs in the miRBase (Release 21.0). In total, 236 conserved miRNAs belonging to 49 families were identified in the nine libraries, with an average of nearly five miRNA members per family (Appendix A). For 24 miRNA families, only one member was found. These conserved miRNAs were aligned to 67 plant species, among which *Glycine max*, *Medicago truncatula*, and *Populus trichocarpa* were the most frequent ones. At the embryogenesis phase, 1,723,573, 1,519,586, and 1,115,971 reads from the three biological replicates perfectly matched 174, 172, and 168 known plant mature miRNAs belonging to 38, 37, and 37 miRNA families, respectively (Table 1). At the seed-filling phase, 777,660, 823,736, and 1,207,854 reads perfectly matched 157, 165, and 183 known plant miRNAs within 38, 41, and 42 families, respectively. At the desiccation phase, 1,230,083, 1,206,643, and 1,271,965 reads perfectly matched 163, 162, and 160 known plant miRNAs within 35, 36, and 34 families, respectively. These conserved miRNAs were combined into 194, 199, and 184 nonredundant conserved miRNAs expressed at the three developmental phases, respectively (Figure 1). Taken together, the number of total reads matching the conserved miRNAs was highest at the embryogenesis phase (4,359,130), followed by those at the desiccation phase (3,708,691) and the seed-filling phase (2,809,250). MIR156 and MIR166 were the two largest miRNA families with 26 and 24 members, respectively (Appendix A). MIR166 was also the most abundantly expressed family at all three phases, followed by MIR167 and MIR396 at the embryogenesis phase, while by MIR159 and MIR167 at the two later phases. There were 18, 21, and 10 conserved miRNAs exclusively expressed at each one of the three developmental phases, respectively (Figure 1). For example, mpi-miR160c-5p and mpi-miR171b-5p were only expressed at the embryogenesis phase, mpi-miR168d-3p and mpi-miR5037 only at the seed-filling phase, and mpi-miR162e-3p and mpi-miR482c-5p only at the desiccation phase. Nevertheless, most of these phase-specific conserved miRNAs were represented by less than 10 reads. On the other hand, 154 conserved miRNAs were expressed throughout all three phases (Figure 1), with their expression levels either remained stable or varied greatly between phases.

The conserved miRNAs identified by homology searching were further subjected to pre-miRNA prediction by exploring the Pongamia seed transcriptome developed by our previous study [15]. The mRNA sequences with hairpin-like structures and with more than 10 miRNA reads anchoring the 5p- and/or 3p-arm were considered as putative pre-miRNAs. As a result, 16 pre-miRNA sequences for the conserved miRNAs were identified, including eight anchored by miRNA reads in the 5p-arm, two by reads in the 3p-arm, and six by reads in both arms (Appendix A). These candidate precursors had a mean length of 149 bp, a GC content of 43.78%, an MFE of –55.56 and an MFEI of –0.90, all of which were conformed to the features of miRNA biogenesis [36].

To predict novel miRNAs in Pongamia seeds, the clean reads excluding those with hits to rRNA, tRNA, snRNA, and snoRNA sequences were aligned with the mRNA sequences in the Pongamia seed transcriptome to retrieve their precursors. Only the reads matching pre-miRNA sequences with characteristic hairpin-like structures and with no homology to previously known plant miRNAs were singled out as novel miRNAs in Pongamia. Overall, there were 143 novel miRNAs identified in Pongamia seeds (Appendix A). The lengths of these novel miRNAs ranged between 20 and 24 nt, with 21 nt being the most common. The majority of these novel miRNAs (135 out of 143) have a precursor anchored by sequencing reads in just one arm. Some novel miRNAs were derived from the same read tag mapping to different unigenes in the Pongamia seed transcriptome. Within the nine libraries, 87, 82, and 68 novel miRNAs were identified from the three biological replicates at the embryogenesis phase, 70, 64, and 83 novel miRNAs were identified at the seed-filling phase, and 97, 93, and 94 novel miRNAs were identified at the desiccation phase (Table 1). These novel miRNAs were combined into 119, 115, and 121 nonredundant novel miRNAs expressed at the three developmental phases, respectively (Figure 1). Compared with the conserved miRNAs, these novel miRNAs were expressed at relatively low levels, the majority of them being represented by less than 10 reads in each library. Fourteen novel miRNAs were specifically expressed at one phase, while 83 novel miRNAs were expressed throughout all three phases (Figure 1). At all three phases, mpi-nmiR0043-5p, mpi-nmiR0124-3p, mpi-nmiR0125-5p, mpi-nmiR0126-3p, and mpi-nmiR0127-5p were the five most abundant novel miRNAs (Appendix A).

### 2.3. Differential Expression of miRNAs during Pongamia Seed Development

To investigate miRNA expression changes during Pongamia seed development, we quantified their expression levels using the transcripts per million (TPM) values, which normalized the read counts of each identified miRNA to the total read counts in each library. Based on the TPM values, we firstly performed Pearson’s correlation analysis and principal component analysis to evaluate the reproducibility of expression data among the three biological replicates. Pearson’s correlation analysis indicated high correlations of miRNA expression levels among replicates within the same developmental phase, with an average coefficient of 0.9991, 0.9980, and 0.9991 for the three phases, respectively (Appendix A). Principal component analysis revealed a clear assignment of three groups corresponding to the three phases (Appendix A), which also demonstrated a good reproducibility of the miRNA expression data yielded in this study.

With a minimum cutoff of 2-fold changes in average TPM values, we sorted out differentially expressed miRNAs (DEmiRs) between developmental phases of Pongamia seeds. A total of 115 DEmiRs, including 82 conserved miRNAs and 33 novel miRNAs, were identified from the two successive comparisons (Appendix A). In the comparison between the embryogenesis phase and the seed-filling phase, 44 significantly up-regulated and 44 significantly down-regulated miRNAs were detected (Figure 2). In the comparison between the seed-filling phase and the desiccation phase, 19 and 39 miRNAs were observed to be significantly up-regulated and down-regulated, respectively. In other words, there were much more DEmiRs (88) between the former two phases than those (58) between the latter two phases.

More specifically, five and eight DEmiRs were significantly up-regulated and down-regulated in both comparisons, respectively (Table 2). Fifteen DEmiRs were significantly up-regulated from the embryogenesis phase to the seed-filling phase, and then significantly down-regulated from the seed-filling phase to the desiccation phase, showing a bell-shape change in expression level (Table 3). On the contrary, three DEmiRs displayed a V-shape change with significant down-regulation followed by significant up-regulation in the two successive comparisons (Table 3). The remaining 84 DEmiRs showed a significant change in expression level only in one comparison. Among those conserved miRNAs, mpi-miR168d-3p and mpi-miR167f-5p showed the highest degree of up-regulation in the former and the latter comparison, respectively, whereas mpi-miR399d-3p and mpi-miR9662a-3p were the most significantly down-regulated ones in the two comparisons, respectively.

### 2.4. Prediction and Functional Annotation of Pongamia miRNA Targets

To gain insight into the regulatory roles of the miRNAs in Pongamia seeds, their putative targets were predicted by aligning the identified miRNA sequences with the unigenes in Pongamia seed transcriptome. As a result, 1327 targets were identified for 210 conserved miRNAs and 121 novel miRNAs through such computational screening, with an average of about four targets per miRNA molecule (Appendix A). Among the conserved miRNAs, mpi-miR156u-5p (40 targets), mpi-miR156j-5p (23 targets), mpi-miR171h-3p (23 targets), mpi-miR156l-5p (22 targets), and mpi-miR396i-3p (22 targets) were the top five with the largest number of target genes in diverse functional categories, implying that they were involved in multiple processes during Pongamia seed development. On the other hand, miRNAs from different families could co-target the same gene.

For instance, Unigene22944 encoding an Auxin Signaling F-Box 2-like protein was co-targeted by four conserved miRNAs (mpi-miR1511-3p, mpi-miR393c-3p, mpi-miR5139-5p, and mpi-miR8155-3p), while Unigene17682 for a disease resistance protein was co-targeted by a conserved miRNA (mpi-miR1510-3p) and a novel miRNA (mpi-nmiR0004-3p). Only 56 targets were identified as transcription factors (TFs) belonging to 23 families, among which NF-YA, bHLH, and B3 were most abundantly represented (Appendix A). Interestingly, we also found eight target genes encoding enzymes in lipid metabolic pathways (Table 4). Among them, only one unigene encoding a mitochondrial cardiolipin synthase was targeted by a conserved miRNA, mpi-miR168f-5p, which was specifically expressed at the desiccation phase. Four unigenes encoding 3-ketoacyl-CoA synthase 4, linoleate 13S-lipoxygenase 3, and phospholipid:diacylglycerol acyltransferase 1, were targeted by the same novel miRNA, mpi-nmiR0017-3p, which expressed only at the embryogenesis and the seed-filling phases in low levels. Unigene22005 encoding a phospholipase A2 was targeted by another novel miRNA, mpi-nmiR0038-5p, with low expressions at the embryogenesis and the desiccation phases. Unigene22800 for a chloroplastic stearoyl-ACP 9-desaturase 6 and Unigene4253 for a malonyl- transferase were targeted by two DEmiRs, mpi-nmiR0028-3p and mpi-nmiR0102-5p, showing a bell-shape pattern of expression and a desiccation-phase-specific expression, respectively.

Among the predicted miRNA targets, 396 genes were targeted by the DEmiRs. This subset of target genes were assigned with GO terms and KEGG pathways, and then subjected to enrichment analysis with the set of unigenes from the whole transcriptome as a background. In the category of Molecular Function, DNA polymerase activity and phosphoric diester hydrolase activity were the two most significantly enriched GO terms (Appendix A). With respect to Biological Process, 48 GO terms were significantly enriched with the DEmiR-targeted genes. Among them, chromosome segregation, regulation of organelle organization, and DNA repair were the most enriched processes. As for Cellular Components, the DEmiR-targeted genes were significantly overrepresented in photosynthetic membrane, clathrin-coated vesicle membrane, and thylakoid membrane. Meanwhile, 74 DEmiR-targeted genes were assigned with 46 KEGG pathways (Appendix A). Three pathways, including non-homologous end-joining, base excision repair, and photosynthesis, were significantly enriched with the DEmiR-targeted genes.

### 2.5. Validation of DEmiRs during Pongamia Seed Development

In order to verify the expression patterns of candidate Pongamia miRNAs obtained from the small RNA sequencing data, we conducted stem-loop qRT-PCR for 15 randomly chosen DEmiRs, including 12 conserved and three novel miRNAs (Appendix A). RNAs extracted from the seeds representing the three developmental phases were used as templates. Note that the RNAs for small RNA sequencing and stem-loop qRT-PCR were separately prepared at the same time points. The stem-loop qRT-PCR results for these 15 DEmiRs were basically consistent with the sequencing results (Figure 3).

Five miRNAs (mpi-miR156c-5p, mpi-miR171b-3p, mpi-miR398c-3p, mpi-miR408a-3p, and mpi-nmiR0028-3p) showed a bell-shape pattern with their expression levels significantly up-regulated from the embryogenesis phase to the seed-filling phase, and then significantly down-regulated from the seed-filling phase to the desiccation phase. Four miRNAs (mpi-miR167c-5p, mpi-miR1507c-3p, mpi-miR2218-3p, and mpi-nmiR0004-3p) were continuously up-regulated during the two successive switches in developmental phase. Conversely, four miRNAs (mpi-miR171j-5p, mpi-miR396c-5p, mpi-miR399b-3p, and mpi-nmiR0135-5p) were down-regulated all through the three phases. The expression level of mpi-miR858a-5p did not change significantly during the first two phases, but then sharply dropped down at the desiccation phase. There was also one miRNA (mpi-miR399d-3p) displaying a near V-shape change in expression level. The linear regression analysis showed a highly significant correlation between the expression profiles revealed by small RNA sequencing and qRT-PCR results (Appendix A). In addition, we also monitored the expression profiles of target genes for these DEmiRs. The DEmiRs with more than three predicted targets were not subjected to such target expression detection. Among six tested DEmiR-targeted genes, five exhibited a V-shape or near V-shape change in expression level (Appendix A). These five target genes included the aforementioned Unigene22800 involved in lipid metabolism, Unigene22770 encoding a superoxide dismutase, Unigene25493 for a cytochrome P450 CYP72A219-like protein, and two unigenes with unknown functions (Appendix A). Besides, a disease resistance protein gene (Unigene8399) was continuously down-regulated as seeds matured. Generally, the tendencies of expression changes for these target genes were opposite to those for the corresponding miRNAs, suggesting that the miRNA-mediated regulation might occur in these targets. The above results again demonstrated that our small RNA sequencing data was reliable in temporal expression analysis of Pongamia miRNAs during seed development.

## 3. Discussion

The seed development process of oilseed plants is instrumental in determining the oil content and quality of the end products. Despite the abundant research on molecular mechanisms regulating seed development at transcriptional level, the miRNA-mediated post-transcriptional regulation of this process remains relatively unstudied, especially in woody oilseed species. In this study, we used high throughput sequencing to characterize miRNAs and examine their expression profiles in developing seeds of Pongamia whose genome has not yet been sequenced, and obtained approximately 283 million small RNA raw reads from nine libraries representing three developmental phases. The small RNAs in Pongamia seeds displayed a wide range of variation in length with the 24-nt RNAs being the most abundant class in all nine libraries, followed by the 21-nt and 22-nt RNAs. Such length distribution pattern of small RNAs was also observed in model plants like Arabidopsis and rice (*Oryza sativa*) [37,38], as well as in some legume relatives (e.g., soybean, *Medicago truncatula*, and peanut) [30,31,39], implying that Pongamia might possess similar processing components for small RNAs biogenesis as other plant species. Moreover, the higher abundance of 24-nt small RNAs was suggested to be related to the silencing of transposons and heterochromatic repeats for ensuring normal seed formation and such abundance of 24-nt small RNAs tended to decrease as seed matured [40], which was also evidently shown in developing Pongamia seeds.

Based on the large quantity of small RNA reads, a total of 236 conserved miRNAs within 49 families and 143 novel miRNAs not found in other plants were identified in this work. The reads matching these miRNAs accounted for less than 6% of the total small RNA reads, indicating that miRNAs only contributed a small portion to small RNAs in Pongamia seeds. Generally, the conserved miRNAs were expressed at higher levels than the novel miRNAs, which was also in accordance with the former results in Arabidopsis and soybean [29,41]. Specifically, 154 conserved and 83 novel miRNAs were expressed at all three phases, implicating that they were indispensable throughout the whole developmental process of Pongamia seeds. The novel miRNAs could be Pongamia-specific miRNAs whose validity needed further confirmation. As for the conserved miRNAs, 15 out of the 49 families (MIR1507, 1510, 1511, 1514, 1515, 2089, 2118, 2119, 2218, 2643, 4403, 482, 5037, 530, and 5559) were only observed in species belonging to the order Fabales. They were classified as Fabales-specific miRNA families and suggested to be young miRNA families by a previous study [29]. Besides, there were also some families appearing in certain species within other eudicot orders (e.g., MIR5139, 8155, and 8175) or monocot orders (e.g., MIR9653, 9662, and 9778) instead of Fabales. These miRNA families probably had an ancient origin, but might have been lost or not yet been identified in other species in Fabales. In other words, both the ancient regulatory pathways and the novel and unique pathways mediated by different kinds of miRNAs might be present in Pongamia.

Identification of the DEmiRs between seed developmental phases and the corresponding target genes could provide valuable information on understanding their functions. In this work, 115 DEmiRs with differential expression in at least one comparison between two successive phases were identified. There were substantially more DEmiRs (88) between the embryogenesis phase and the seed-filling phase than those (58) between the seed-filling phase and the desiccation phase. Consistent with this tendency, more protein-coding genes were observed to be differentially expressed between the former two phases than those between the latter two phases [35]. To some extent, the general expression patterns of both DEmiRs and differentially expressed genes (DEGs) correlated with a more remarkable change in seed traits and oil content during the early developmental stages of Pongamia seeds [35,42]. Meanwhile, there were substantially more down-regulated DEmiRs (39) than up-regulated ones (19) from the seed-filling phase to the desiccation phase. Considering that miRNAs always negatively regulate protein-coding genes, the above observation also coincides with our previous findings of more up-regulated genes than down-regulated ones during the same developmental switch [35]. The DEmiRs identified in this study could potentially target 396 genes, which were preferentially enriched in certain GO terms or KEGG pathways associated with DNA repair and photosynthesis. DNA damage repair was crucial for the maintenance of genome integrity over cell division during seed development [43], while photosynthesis could provide energy and oxygen for enhancing biosynthetic fluxes to lipids in developing legume seeds [44]. Therefore, the enrichment of DEmiR-targeted genes in these two functional categories was quite reasonable.

Involvements of miRNAs in seed development have already been established in several conserved miRNA families. By binding to plant-specific transcription factor genes *SPL10* and *SPL11*, miR156 could negatively regulate phase transition and seed maturation while its expression gradually declined as seed developed [25]. Some members of the MIR156 family could also target certain genes in fatty acid biosynthetic pathway and could thus affect seed oil content of rapeseed [45]. In this study, nine MIR156 members were screened out as DEmiRs and they targeted dozens of genes with diverse functions (Appendix A). Since both *SPL10* and *SPL11* transcripts were not found in the Pongamia seed transcriptome [15], none of those nine members were predicted to target any *SPL* homologs. Notably, mpi-miR156a-3p and mpi-miR156c-5p displayed significant changes in expression levels in both comparisons between seed developmental phases. The former targeted a protein S-acyltransferase (PAT) 10 for protein lipid modification and was continuously down-regulated from the embryogenesis phase to the desiccation phase, whereas the latter targeted a cytochrome P450 CYP72A219-like protein for metal binding and showed a bell-shape pattern of expression changes. The expression changes of mpi-miR156c-5p and its target gene (Unigene25493) were further validated by qRT-PCR. These two miRNAs might play critical roles at the early and mid stages of seed development in Pongamia as those reported in some other plants [25,45].

As another upstream regulator of seed development, miR160 could repress *ARF10* and *ARF17* to modulate the expression of early auxin response genes [26,46]. In this study, mpi-miR160c-5p was exclusively expressed at the embryogenesis phase and thus recognized as a DEmiR in developing Pongamia seeds. The predicted targets of this miRNA were *ARF17* and *ARF18*. This result implied that miR160 might mainly function at early developmental stages of Pongamia seeds.

The miR164-mediated regulation of the No Apical Meristem (NAC) transcription factors also played an essential role in seed development. Expression of a miR164-resistant version of Cup-Shaped Cotyledon (CUC) 1 within the NAC family could cause cotyledon orientation defects [47]. Among the three MIR164 members identified in this work, only mpi-miR164a-5p was a DEmiR with significant down-regulation from the embryogenesis phase to the desiccation phase. This miRNA was predicted to target the transcripts of a HHE cation-binding domain protein and a UPF0481 protein instead of any NAC domain-containing proteins in the Pongamia seed transcriptome, suggesting the existence of specialized regulatory components coupled with miR164 in this species, which was different from the miR164-NAC module in other plants.

The MYB transcription factors usually serve as positive regulators of ABA responses. By silencing *MYB33* and *MYB101*, miR159 could act as a negative regulator preventing the transition from seed dormancy to germination [48]. In addition, the miR159-mediated regulation of MYB33 and MYB65 expression might also be responsible for determining seed size [49]. We found four MIR159 members in the list of DEmiRs (Appendix A). Except for one member showing a bell-shape expression pattern, the other three members (mpi-miR159a-5p, mpi-miR159b-3p, and mpi-miR159d-5p) all exhibited a significant down-regulated expression in favor of releasing the constraints on seed germination.

The APETALA2 (AP2) transcription factors compose a large family with a variety of regulatory functions, including the control of seed size and seed mass [50,51]. Although the *AP2* genes could be regulated by miR172 [52], there was still no evidence supporting the direct involvement of the miR172-AP2 module in seed development. Additionally, up-regulation of miR172 was shown to promote seed germination through its interaction with miR156 and the *Delay of Gemination1* (*DOG1*) gene [53]. However, mpi-miR172c-3p was significantly down-regulated as seed matured and it could target diverse transcripts including an *AP2* homolog in Pongamia (Appendix A). Hence, we speculated that the miR172 might participate in the regulation of seed development via diversified mechanisms in different plant species.

In addition to the above conserved miRNAs having been reported to be relevant to seed development, there are also some other DEmiRs which may be candidate regulators for this process in Pongamia. For instance, mpi-miR482a-3p displayed a significantly increasing expression and could target several transcripts encoding a tyrosyl-DNA phosphodiesterase (TDP) 1 for DNA 3′-end processing [54]. As aforementioned, DNA damage repair was crucial for maintaining genome integrity during seed development. Another interesting candidate was mpi-nmiR0028-3p which targeted a stearoyl-ACP 9-desaturase 6 for converting stearic acid to oleic acid. This novel miRNA was also a DEmiR with a bell-shape expression pattern and could probably make an impact on fatty acid composition in developing Pongamia seeds. The contrasting expression changes of this novel miRNA and its target gene (Unigene22800) were also verified by qRT-PCR (Figure 3 and Appendix A).

## 4. Materials and Methods

### 4.1. Plant Material and Sample Collection

Three eight-year-old Pongamia trees cultivated in Shenzhen University, Shenzhen, China (22°32′ N, 113°55′ E), with the monthly average temperature ranging from about 15°C to 29°C and an average annual rainfall of approximately 2000 mm, were used as the biological replicates for seed sampling. The inflorescences on different sub-branches of each tree were labeled with tags recording their first flowering dates. Pods were harvested at 10 weeks after flowering (WAF), 20 WAF, and 30 WAF, representing the three developmental phases of Pongamia seeds as previously described [35]. At each sampling time point, the seeds were manually separated from pods, washed with distilled water, immediately frozen in liquid nitrogen, and then stored at −80°C before RNA extraction for further experiments.

### 4.2. RNA Isolation, Library Construction and Small RNA Sequencing

Total RNA was isolated from Pongamia seeds using a modified CTAB method [55]. At each sampling time point, the seeds from each of the three trees were separately subjected to RNA extraction. The resulting nine RNA samples were further purified with the RNeasy Plant Mini Kit (Qiagen, Hilden, Germany) following the manufacturer’s instructions. RNA concentration and quality of each sample was assessed by an Agilent 2100 Bioanalyzer (Agilent Technologies, Palo Alto, CA, USA). Subsequently, the RNA molecules with different sizes were separated using polyacrylamide gel electrophoresis. The 18–30 nt small RNAs were excised and then ligated to 5′ and 3′ adapters by T4 RNA ligase in two separate steps. The ligation products were reverse transcribed and then amplified by PCR. The 140–160 bp PCR products were enriched to generate nine cDNA libraries for sequencing on the Illumina HiSeq 2500 platform (Illumina, San Diego, CA, USA) at GeneDenovo Biotechnology Co., Ltd. (GeneDenovo, Guangzhou, China). All sequence data of the nine libraries were deposited in the NCBI Sequence Read Archive (SRA) database under the accession number PRJNA550227.

### 4.3. Sequencing Data Processing and Identification of Conserved and Novel miRNAs

Raw reads of the nine libraries were firstly filtered by in-house Perl scripts to remove low-quality reads, reads containing 5′ adapters and polyA tails, reads without 3′ adapters, and reads shorter than 18 nt. Then, the resulting clean reads were blasted against the NCBI GenBank (http://www.ncbi.nlm.nih.gov/genbank/) database and the Rfam (http://rfam.xfam.org/) database to screen out rRNAs, tRNAs, snRNA, and snoRNAs. The remaining reads without matches to the above four types of RNAs were searched against the miRBase (21.0) (http://www.mirbase.org/) database to identify phylogenetically conserved miRNAs. Only the reads with no mismatches to those currently known plant miRNA sequences were considered as conserved miRNAs. The remaining unannotated reads were aligned with the Pongamia seed transcriptome to predict potential novel miRNAs and their hairpin precursors by the Mireap software (http://sourceforge.net/projects/mireap/) with default settings. The miRNA expression levels were calculated and normalized as transcripts per million (TPM) values based on the total number of clean reads in each library. For miRNAs that were not expressed in the samples, the expression levels were set to 0.01. To determine the statistical significance of expression difference, the fold change and the *P* value were calculated for each pairwise comparison between seed developmental phases. Both absolute fold change ≥ 2 and *P* value ≤ 0.05 were adopted as the thresholds to identify DEmiRs.

### 4.4. Identification and Analysis of miRNA Targets

Putative target genes of conserved and novel miRNAs were predicted by the PatMatch software (https://www.arabidopsis.org/cgi-bin/patmatch/nph-patmatch.pl), which queried the Pongamia seed transcriptome with the parameters as follows: no more than four mismatches between miRNA and target (G-U bases count as 0.5 mismatches); no more than two adjacent mismatches in the miRNA/target duplex; no adjacent mismatches in positions 2–12, no mismatches in positions 10–11, and no more than 2.5 mismatches in positions 1–12 of the miRNA/target duplex (5′ of miRNA); and the minimum free energy (MFE) of the miRNA/target duplex should be ≥ 75% compared to the MFE of the miRNA bound to its perfect complement. The predicted target genes were assigned with GO (http://www.geneontology.org/) and KEGG (http://www.genome.jp/kegg/) annotations. Specifically, for those genes targeted by DEmiRs, the hypergeometric tests were conducted to identify significantly enriched GO terms or KEGG pathways with all unigenes in the Pongamia seed transcriptome as a background. The calculated *P* values were gone through false discovery rate (FDR) correction, taking FDR ≤ 0.05 as a threshold.

### 4.5. Quantitative Real-Time PCR for miRNAs and Target Genes

To validate changes in expression levels of miRNAs during the three developmental phases of Pongamia seeds, total RNAs were isolated and enriched for small RNA fractions as above described, and then reverse transcribed by stem-loop RT-PCR [56]. The resulting cDNAs were diluted and subjected to quantitative real-time PCRs on an ABI PRISM 7300 Sequence Detection System (Applied Biosystems, Foster City, CA, USA) using the ChamQ Universal SYBR qPCR Master Mix (Vazyme Biotech, Nanjing, China). Primers for stem-loop reverse transcription and real-time PCR were separately designed for 15 randomly selected DEmiRs (Appendix A). For expression profiling of target genes, total RNAs from the same samples as those for miRNA profiling were employed for first-strand cDNA synthesis using the Super Script First-Strand cDNA Synthesis Kit (Invitrogen, Carlsbad, CA, USA), followed by Real-Time PCR with specific primers for target genes (Appendix A). U6 snRNA and actin were used as the internal reference for the normalization of miRNAs and target genes, respectively. All real-time PCRs were run in three technical replicates. The primer specificity was confirmed by melting curve analysis. The relative expression levels of the tested miRNAs and target genes were calculated with the 2^−ΔΔCt^ method.

## 5. Conclusions

In summary, we presented the first large-scale collection of small RNAs from Pongamia seeds by high-throughput sequencing and identified 236 conserved miRNAs belonging to 49 families as well as 143 novel miRNAs. Among them, 82 conserved miRNAs and 33 novel miRNAs with significantly differential expression between successive developmental phases were sorted out as DEmiRs. The enrichment of DEmiR-targeted genes in the functional categories related to DNA damage repair and photosynthesis suggested that the regulation of these two processes should be pivotal in controlling Pongamia seed development. Through the combined analyses of expression profiles for DEmiRs and functional annotations for their target genes, we not only found some miRNAs within the families (MIR156, MIR159, MIR160, MIR164, and MIR172) previously confirmed to be capable of regulating seed development in herbaceous model plants, but also proposed some candidates (e.g., mpi-miR482a-3p and mpi-nmiR0028-3p) with specialized regulatory mechanisms for seed development and fatty acid biosynthesis in this woody plant. These miRNA candidates deserve further functional characterization and may be potentially applied in genetic breeding of new varieties with desired seed traits for Pongamia and other oilseed plants.

## Figures and Tables

**Figure 1 ijms-20-03509-f001:**
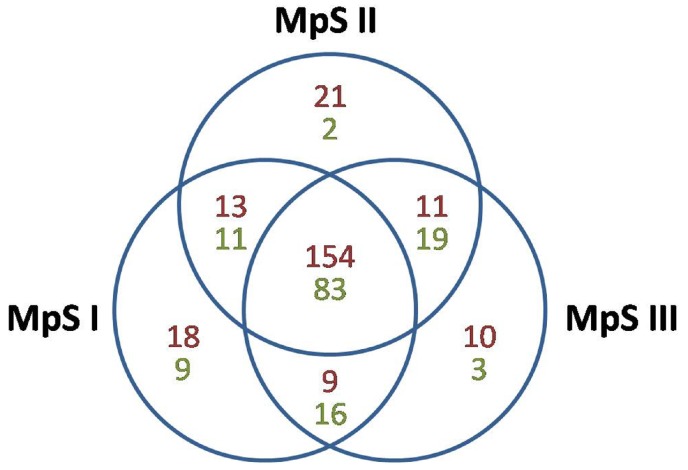
Venn diagram showing the distribution of miRNAs among the three developmental phases of Pongamia seeds. MpSI, the embryogenesis phase; MpSII, the seed-filling phase; MpSIII, the desiccation phase. The values in red and green indicate the numbers of conserved and novel miRNAs, respectively.

**Figure 2 ijms-20-03509-f002:**
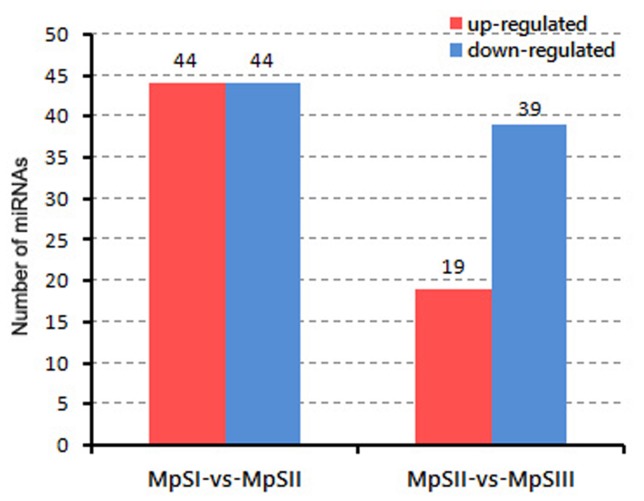
The numbers of DEmiRs between developmental phases of Pongamia seeds. The values in red and blue indicate the numbers of up-regulated and down-regulated miRNAs, respectively.

**Figure 3 ijms-20-03509-f003:**
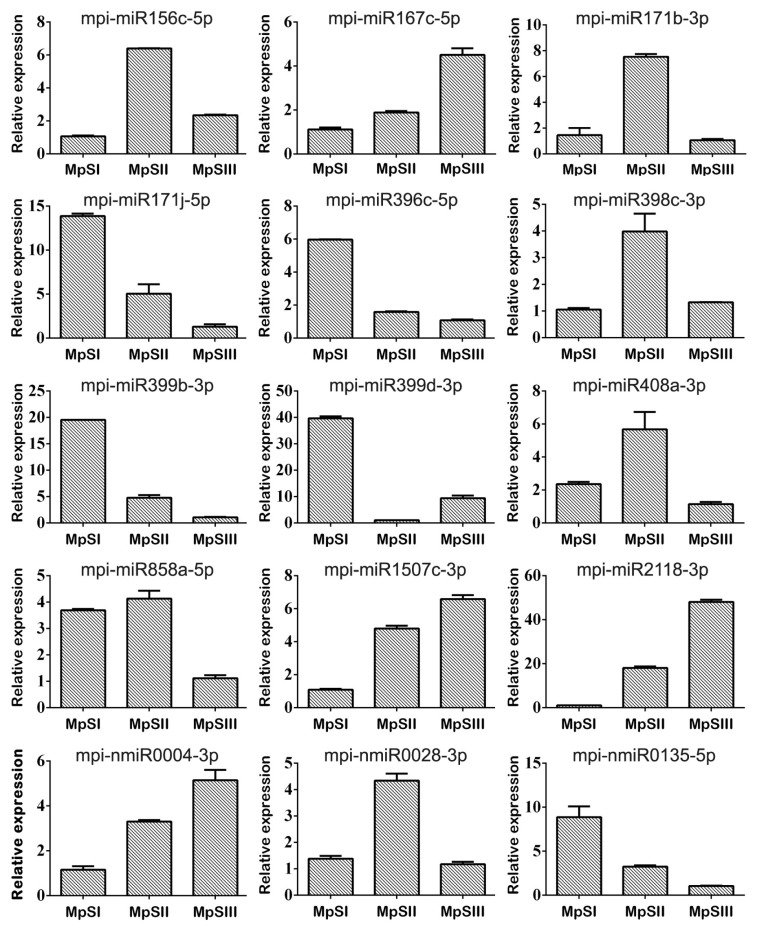
Relative expression levels of 15 DEmiRs evaluated by stem-loop qRT-PCR. An U6 snRNA gene of Pongamia was used as internal control. Bars represent standard deviations of three technical replicates.

**Table 1 ijms-20-03509-t001:** Statistics of sequencing reads and miRNAs.

Statistical Items	MpSI-1	MpSI-2	MpSI-3	MpSII-1	MpSII-2	MpSII-3	MpSIII-1	MpSIII-2	MpSIII-3
Number of raw reads	31,711,290	31,959,711	31,763,941	27,733,326	33,988,872	31,143,787	31,167,480	32,681,945	31,100,220
Number of clean reads	21,400,622	21,401,693	21,217,533	17,612,476	21,414,373	19,377,920	20,039,686	20,872,904	19,792,612
Retention rate	67.49%	66.96%	66.80%	63.51%	63.00%	62.22%	64.30%	63.87%	63.64%
Number of unique tags	2,757,270	2,876,894	2,882,107	3,066,337	3,427,062	2,942,567	2,593,393	2,774,500	2,443,701
Number of conserved miRNAs	174	172	168	157	165	183	163	162	160
Number of novel miRNAs	87	82	68	70	64	83	97	93	94

**Table 2 ijms-20-03509-t002:** Pongamia miRNAs with the same tendencies of expression changes in two comparisons between developmental phases.

miRNA_ID	Reads Count									Fold Change	
	MpSI-1	MpSI-2	MpSI-3	MpSII-1	MpSII-2	MpSII-3	MpSIII-1	MpSIII-2	MpSIII-3	MpSII/MpSI	MpSIII/MpSII
mpi-miR156f-5p	0	0	0	2	2	4	12	13	12	7.65	1.74
mpi-miR482a-3p	2380	4221	2288	5869	5817	7435	39410	19325	18703	1.58	1.49
mpi-nmiR0004-3p	21	20	9	25	22	24	69	80	106	1.07	1.28
mpi-nmiR0083-3p	0	0	0	11	11	9	71	70	84	9.71	2.28
mpi-nmiR0119-5p	0	0	0	2	2	6	10	8	14	7.92	1.26
mpi-miR156a-3p	2112	3098	1195	736	356	577	316	347	424	−1.39	−1.19
mpi-miR164a-5p	141	121	130	7	20	32	11	7	7	−2.37	−1.67
mpi-miR166a-5p	3344	2597	2102	718	822	1330	437	495	803	−1.03	−1.22
mpi-miR166k-5p	408	297	290	19	21	30	7	4	9	−3.37	−2.32
mpi-miR171j-5p	66	54	35	5	5	5	0	2	0	−2.83	−3.43
mpi-miR393b-5p	384	272	219	44	84	133	35	38	42	−1.31	−1.64
mpi-miR399a-3p	5907	4743	3838	908	1026	1399	725	552	741	−1.65	−1.24
mpi-miR399b-3p	3876	4196	2814	1094	848	1048	412	207	306	−1.35	−2.25

**Table 3 ijms-20-03509-t003:** Pongamia miRNAs with the opposite tendencies of expression changes in two comparisons between developmental phases.

miRNA_ID	Reads Count									Fold Change	
	MpSI-1	MpSI-2	MpSI-3	MpSII-1	MpSII-2	MpSII-3	MpSIII-1	MpSIII-2	MpSIII-3	MpSII/MpSI	MpSIII/MpSII
mpi-miR156c-5p	84	111	69	327	203	197	95	68	83	2.01	−2.16
mpi-miR168d-3p	0	0	0	2	4	9	0	0	0	8.49	−8.49
mpi-miR171b-3p	96	94	52	170	226	432	219	142	198	2.22	−1.02
mpi-miR398c-3p	393	398	260	964	1092	1553	360	317	436	2.25	−2.21
mpi-miR398f-3p	10	10	6	113	142	169	9	7	13	4.52	−4.40
mpi-miR408a-3p	130	137	75	379	446	612	176	170	199	2.56	−1.91
mpi-miR408b-3p	15	15	6	68	98	147	35	45	59	3.62	−1.66
mpi-miR408d-3p	7	8	8	84	63	76	37	54	45	3.75	−1.28
mpi-miR4403-5p	3	3	3	6	5	6	2	2	2	1.39	−2.06
mpi-miR5037	0	0	0	2	2	2	0	0	0	7.32	−7.32
mpi-nmiR0028-3p	7	6	7	25	32	19	7	7	3	2.44	−2.75
mpi-nmiR0034-3p	4	5	3	6	6	9	4	2	3	1.28	−1.73
mpi-nmiR0050-5p	0	0	0	5	3	6	0	0	0	8.50	−8.50
mpi-nmiR0072-5p	14	25	12	16	28	47	2	8	5	1.25	−3.06
mpi-nmiR0103-5p	0	0	0	4	16	10	0	0	0	9.63	−9.63
mpi-miR396c-3p	1094	422	725	69	78	151	401	240	257	−2.50	1.13
mpi-miR396e-5p	9	8	4	0	0	0	5	3	5	−8.57	7.88
mpi-miR399d-3p	43	33	29	0	0	0	5	7	2	−10.94	7.99

**Table 4 ijms-20-03509-t004:** Pongamia miRNAs with putative targets in relation to lipid metabolism.

miRNA_ID	Target_ID	Target_Annotation	MpSI_TPM	MpSII_TPM	MpSIII_TPM
mpi-miR168f-5p	Unigene24517	cardiolipin synthase (CMP-forming), mitochondrial	0.01	0.01	0.36
mpi-nmiR0017-3p	Unigene10960	3-ketoacyl-CoA synthase 4	1.02	1.02	0.01
	Unigene15710	linoleate 13S-lipoxygenase 3-1, chloroplastic			
	Unigene26514	phospholipid:diacylglycerol acyltransferase 1-like			
	Unigene49632	3-ketoacyl-CoA synthase 4			
mpi-nmiR0028-3p	Unigene22800	stearoyl-ACP 9-desaturase 6, chloroplastic	11.54	20.81	3.09
mpi-nmiR0038-5p	Unigene22005	phospholipase A2	1.42	0.01	0.37
mpi-nmiR0102-5p	Unigene4253	malonyltransferase	0.01	0.01	2.53

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
