# Peer review of "Unravelling the MicroRNA-Mediated Gene Regulation in Developing Pongamia Seeds by High-Throughput Small RNA Profiling"

_ijms, 2019, doi:10.3390/ijms20143509_

Round 1

Reviewer 1 Report

The paper by Jin et al. represents a step forward in the characterization of the transcriptional landscape in Pongamia trees, a plant with biochemical and ecological properties that make it suitable for the biofuel industry. The authors present the first large-scale collection of small RNAs from Pongamia seeds, identify conserved, novel and differentially expressed (DE) miRNAs between three developmental stages, as well as enriched DEmiRNA-targeted genes, and validate observed expression changes through qRT PCR, thus providing valuable information for potential miRNA candidates likely to be tested in future genetic breeding of new Pongamia varieties.

Overall, the paper is well written, the interest and aims are clearly stated, methodologies and statistical analyses are appropriate, results are sound and clearly presented, and the discussion is up-to-date and comprehensive.

A few minor comments:

In Table 1; are the numbers of novel miRNAs for the raw reads or the clean reads? and if yes, do they include or not rRNA, tRNA, snRNA, and snoRNA sequences? how these relates to those numbers of Figure 1?

Line 117: the observed differences between developmental phases are different but doesn´t seem to be significant. Perhaps a clarification would help to place your statement in context.

Lines 279-280: such implication read a bit speculative...is that necessary?

In the Discussion, I couldn´t find a sentence with the authors´ viewpoint regarding the observed changes along the developmental phases analyzed. During seed maturation (and desiccation) I would expect that many genes are sequentially shut down.  How the observed DE tendencies along the seed development in Pongamia are, compared to what would be expected from the process of seed maturation (and desiccation)?

Author Response

1)      We thank the reviewer for kindly reminding us to delineate the difference between the numbers of novel miRNAs in Table 1 and those in Figure 1. As we described in the Materials and Methods section 4.3, the clean reads were first subjected to the removal of rRNA, tRNA, snRNA, and snoRNA sequences, and then to the filtration of conserved miRNA sequences, and finally to the prediction of novel miRNA sequences. In other words, the novel miRNA sequences are a subset of the clean reads not including the rRNA, tRNA, snRNA, and snoRNA sequences.

The green numbers in Figure 1 indicate the numbers of novel miRNAs with matching reads in at least one biological replicate at each developmental phase. For example, we found 87, 82, and 68 novel miRNAs from the three biological replicates at the MpSI phase (Line 155 and Table 1). These novel miRNAs were combined into 119 nonredundant novel miRNAs expressed at the MpSI phase as shown in Figure 1, including 9 MpSI-specific novel miRNAs, 11 novel miRNAs shared by MpSI and MpSII, 16 novel miRNAs shared by MpSI and MpSIII, and 83 novel miRNAs shared by all three phases.

Following the reviewer’s comments, we added a sentence ‘These novel miRNAs were combined into 119, 115, and 121 nonredundant novel miRNAs expressed at the three developmental phases, respectively (Figure 1).’ in Line 157-159. Accordingly, we also added a sentence ‘These conserved miRNAs were combined into 194, 199, and 184 nonredundant conserved miRNAs expressed at the three developmental phases, respectively (Figure 1).’ in Line 117-119 to explain the numbers of conserved miRNAs in Table 1 and Figure 1.

2)      To clarify the differences between developmental phases, we totalized the matching reads at the three developmental phases and rewrote the sentence as ‘Taken together, the number of total reads matching the conserved miRNAs was highest at the embryogenesis phase (4,359,130), followed by those at the desiccation phase (3,708,691) and the seed-filling phase (2,809,250).’ in Line 119-122.

3)      ‘The higher abundance of 24-nt small RNAs was suggested to be related to the silencing of transposons and heterochromatic repeats for ensuring normal seed formation and such abundance of 24-nt small RNAs tended to decrease as seed matured [40].’ – This sentence, quoted from Zabala et al. (2012), helps to support our observation that the 24-nt small RNAs are prominent in developing Pongamia seeds as observed in other plants. It also helps us to explain the correlation between the abundance of 24-nt small RNAs and the process of seed maturation. We tend to retain this sentence in the text.

4)      Again, we thank the reviewer for this helpful suggestion on describing DEmiRs tendencies along the seed development in Pongamia. Accordingly, we added two sentences in the Discussion section Line 318-322 as follows. ‘Meanwhile, there were substantially more down-regulated DEmiRs (39) than up-regulated ones (19) from the seed-filling phase to the desiccation phase. Considering that miRNAs always negatively regulate protein-coding genes, the above observation also coincides with our previous findings of more up-regulated genes than down-regulated ones during the same developmental switch [35].’

Reviewer 2 Report

Jin et al., reported an interesting manuscript on the miRNA candidates for further functional characterization and breeding practice in Pongamia and other oilseed plants. The manuscript is well written and easy to follow. I recommend the manuscript for publication in Molecular Sciences. However, before, I suggest recommending making the following changes:

-          Line 36. Please add a description of which environment.

-          Line 36. The scientific name (Pongamia) can be avoided. The authors already reported the scientific name in the abstract.

-          Sometime the authors did not report the scientific name of the mentioned crops. Please report the scientific name for each mentioned crop in the manuscript.

-          Line 42. Please add a description of the main pharmacological activities

-          In the subsection 4.1 the authors should report more information on the agronomic management and the parameters regarding the weather conditions such as Tmax, Tmin, and rainfall (mm)

Author Response

1)      We added the words ‘such as mountain area or intertidal zone’ in Line 36-37.

2)      We deleted the scientific name of Pongamia in Line 37.

3)      We thank the reviewer for reminding us the use of scientific names for all crops in the manuscript. We added the scientific names for soybean, rapeseed, and maize in Line 34-35, for peanut in Line 68, and for rice in Line 284.

4)      We added several main pharmacological activities, including antioxidant, antimicrobial, anti-diabetic, and anti-hyperammonemic in Line 43-44.

5)      We added a description of the weather conditions in Line 389-390.